# Enantioselective Synthesis and Pharmacological Evaluation of Aza-CGP37157–Lipoic Acid Hybrids for the Treatment of Alzheimer’s Disease

**DOI:** 10.3390/antiox11010112

**Published:** 2022-01-04

**Authors:** Ángel Cores, Patrycja Michalska, José Miguel Pérez, Enrique Crisman, Clara Gómez, Mercedes Villacampa, José Carlos Menéndez, Rafael León

**Affiliations:** 1Unidad de Química Orgánica y Farmacéutica, Departamento de Química en Ciencias Farmacéuticas, Facultad de Farmacia, Universidad Complutense, 28040 Madrid, Spain; acores@ucm.es (Á.C.); josemigp@ucm.es (J.M.P.); mvsanz@ucm.es (M.V.); 2Department of Chemistry, Imperial College London, London SW7 2BX, UK; p.dziama@imperial.ac.uk; 3Instituto de Investigación Sanitaria, Hospital Universitario de la Princesa, 28006 Madrid, Spain; ecrisman@outlook.com; 4Instituto de Química Médica, Consejo Superior de Investigaciones Científicas (IQM-CSIC), 28006 Madrid, Spain; clara.gomezserrano@estudiante.uam.es

**Keywords:** CGP37157, lipoic acid, neuroprotection, NRF2 induction

## Abstract

Hybrids based on an aza-analogue of CGP37157, a mitochondrial Na^+^/Ca^2+^ exchanger antagonist, and lipoic acid were obtained in order to combine in a single molecule the antioxidant and NRF2 induction properties of lipoic acid and the neuroprotective activity of CGP37157. The four possible enantiomers of the hybrid structure were synthesized by using as the key step a fully diastereoselective reduction induced by Ellman’s chiral auxiliary. After computational druggability studies that predicted good ADME profiles and blood–brain permeation for all compounds, the DPPH assay showed moderate oxidant scavenger capacity. Following a cytotoxicity evaluation that proved the compounds to be non-neurotoxic at the concentrations tested, they were assayed for NRF2 induction capacity and for anti-inflammatory properties and measured by their ability to inhibit nitrite production in the lipopolysaccharide-stimulated BV2 microglial cell model. Moreover, the compounds were studied for their neuroprotective effect in a model of oxidative stress achieved by treatment of SH-SY5Y neuroblastoma cells with the rotenone–oligomycin combination and also in a model of hyperphosphorylation induced by treatment with okadaic acid. The stereocenter configuration showed a critical influence in NRF2 induction properties, and also in the neuroprotection against oxidative stress experiment, leading to the identification of the compound with *S* and *R* configuration as an interesting hit with a good neuroprotective profile against oxidative stress and hyperphosphorylation, together with a relevant anti-neuroinflammatory activity. This interesting multitarget profile will be further characterized in future work.

## 1. Introduction

The rise in life expectancy is associated with an increase in the prevalence of age-related pathologies, which are becoming one of the most important socioeconomic challenges of the near future [1]. Among age-related diseases, neurodegenerative diseases (NDDs) such as Parkinson’s disease (PD), stroke, amyotrophic lateral sclerosis (ALS) or Alzheimer’s disease (AD) are characterized by having a multifactorial origin in which genetic and environmental factors are combined to elicit the pathology. NDDs’ common hallmarks include protein misfolding [2], ion homeostasis and metabolic imbalance [3,4], neuroinflammation [5], mitochondrial dysfunction [6] and oxidative stress [7]. Frequently, these factors are interconnected in complex networks, meaning that tackling a single target would not be possible to reverse the pathological condition [5].

The nuclear factor erythroid 2-related factor 2 (NRF2) is a major regulator of cellular redox homeostasis, and the activation of this pathway provides an effective mechanism against oxidative or electrophilic stress. NRF2 is bound to Kelch-like ECH-associated protein 1 (KEAP1) in the cytosol under physiological conditions. KEAP1 acts as an E3 ligase recruiter by tagging NRF2 with ubiquitin and thus promoting its proteasomal degradation. Thus, NRF2 is constitutively synthesized and rapidly degraded by the proteasome under basal conditions. However, electrophilic or oxidative stimuli cause chemical modifications in a cysteine-rich region in KEAP1 to induce a conformational change, preventing NRF2 proteasomal degradation. Then, NRF2 translocates into the nucleus and binds the antioxidant response element (ARE) sequences of DNA to increase the transcription of cytoprotective and antioxidant genes [8].

Oxidative stress (OS) is a major and shared cause in all NDDs. Neuronal tissues are sensitive to oxidative damage due to their high content in groups sensitive to oxidation and their limited regenerative capacity [5]. OS is caused by an imbalance between the increased production of reactive oxygen or nitrogen species and the decrease in the antioxidant defenses of the cells. Furthermore, OS may be related to other causes, such as mitochondrial dysfunction, which is intensified during aging. During such processes, there is an uncontrolled increase in sodium mediated by the mitochondrial Na^+^/Ca^2+^ exchanger (NCLX), which reduces the mobility of ubiquinone between complexes II and III of the electron transport chain, and it produces large amounts of hydroperoxide species [9].

A multitude of compounds with antioxidant activity have shown their usefulness in in vitro models of neurodegeneration, both of synthetic and natural sources. An example of a naturally occurring antioxidant is lipoic acid, which acts as a cofactor for α-ketoacid reductase in the mitochondria and plays a key role in the control of mitochondrial energy metabolism [10]. In addition to this activity as an antioxidant, lipoic acid has been shown to induce NRF2, promoting the transcription of cytoprotective proteins and thus contributing to its antioxidant and anti-inflammatory activity [11]. A similar pharmacological profile is present in the NCLX-blocking 1,5-benzodiazepine (CGP37157) [12], which has been shown to be a useful neuroprotective agent [13] acting through the reduction of reactive oxygen species in the mitochondria after oxygen deprivation [8].

In addition to OS, glial over-activation and neuroinflammation contribute to neuronal demise in NDDs. Glial cells constitute the innate immune system of the CNS and are essential to maintain central nervous system (CNS) homeostasis [14]. Glial cells express receptors that can sense different stimuli, and once activated, they acquire an inflammatory phenotype and release numerous cytokines and chemokines, including TNFα and IL-1β [15]. Moreover, they further contribute to OS through the activation of the inducible nitric oxide sintase (iNOS) and NADPH oxidase (NOX) enzymes, which release nitric oxide and superoxide, respectively [16]. In NDDs, the presence of OS and characteristic protein aggregates sustain a continuous microglial activation that becomes deleterious and promotes neuronal death. For example, it is known that amyloid-β (Aβ) aggregates present in AD brains activate the toll-like receptors (TLRs) on glial cells, hence increasing the pro-inflammatory mediators and toxicity [17]; moreover, OS impairs the response of redox sensitive transcription factors, such as NF-KB and NRF2, leading to a chronic glial inflammation [18]. Finally, dying neurons release ATP and free radicals, which can further activate purinergic receptors on glial cells sustaining feedback loops that contribute to neurodegeneration.

Based on its antioxidant profile, our group synthesized derivatives an aza-CGP37157–lipoic acid hybrid (Figure 1) that showed a promising profile as a neuroprotective agent due to the induction of NRF2 and antioxidant activity [19]. However, the study of these compounds had an important limitation: they were studied as racemic mixtures, since it was not possible to obtain the aza-CGP-37157 derivative in the form of a single enantiomer.

In this article we describe the synthesis of aza-CGP37157–lipoic hybrids as single enantiomers, obtained by means of a diastereoselective reduction that has made it possible to obtain both pure enantiomers of aza-CGP37157 and subsequently combine them with both enantiomers of lipoic acid. The pharmacological evaluation of these enantiomerically pure hybrids has made it possible to identify the absolute configuration associated to the best profile as a neuroprotective and as NRF2 inducer. Some parts of the present work come from a doctoral thesis [20].

## 2. Materials and Methods

### 2.1. Chemistry

#### 2.1.1. General Experimental Details

All reagents and solvents were of commercial quality and were used as received. Reactions were monitored by thin-layer chromatography on aluminum plates coated with silica gel and fluorescent indicator. Microwave-assisted reactions were performed on a CEM Discover focused microwave reactor. Separations by flash chromatography were performed by using a Combiflash Teledyne automated flash chromatograph or on conventional silica gel columns. Melting points were measured with a Kofler-type heating platine microscope from Reichert, 723 model, and are uncorrected. Infrared spectra were recorded with an Agilent Cary630 FTIR spectrophotometer with a diamond ATR accessory for solid and liquid samples, requiring no sample preparation; wavenumbers are given in cm^−1^. NMR spectroscopic data were obtained by using spectrometers maintained by the CAI de Resonancia Magnética, UCM, operating at 250, 300, 500 or 700 MHz for ^1^H NMR and 63, 75, 100 or 176 MHz for ^13^C NMR; chemical shifts are given in (δ) parts per million and coupling constants (*J*) in hertz. Elemental analyses were determined by the CAI de Microanálisis, Universidad Complutense, using a Leco CHNS-932 combustion microanalyzer. The analysis of enantiomeric purity was conducted in an HPLC Agilent 1220 Infinity LC chromatograph with a chiral column ULTRON ES OVM Chiral Analytical Reverse Phase, 5 μm, 4.6 mm × 150 mm

#### 2.1.2. Synthesis of Sulfinilimine **1**

First, 2-Amino-2′,5-dichlorobenzophenone (1.0 g, 8 mmol), (*R*)- or (*S*)-2-methyl-2-propanesulfinamide (1.0 eq.) and neat Ti(OEt)_4_ (4.0 eq.) were placed in a sealed tube under an argon steam. Then, the mixture was heated at 100 °C for 1 h under microwave irradiation. After cooling the reaction, a mixture of water (5 mL) and ethyl acetate (10 mL) was added, and the resulting slurry was filtered through a Celite plug and washed gently with ethyl acetate. The organic layer from the filtrate was washed with brine, dried over anhydrous Na_2_SO_4_ and the solvent was removed in vacuo. The residue was purified by column chromatography on silica gel, using as eluent a 6:4 n-hexane/ethyl acetate mixture to afford 1.24 g (42%) of **1** as a yellow solid.

##### (*R*,*Z*)-*N*-((2-Amino-5-chlorophenyl)(2-chlorophenyl)methylene)-2-methylpropane-2-sulfinamide (***R*-1**)

mp: 138–139 °C; [α]${}_{D}^{25}$ = −40.00 (c = 1.00 mg/mL, CHCl_3_); IR (neat, cm^−1^): 3373, 3273, 1516, 1237, 1053; ^1^H NMR (250 MHz, CDCl_3_) δ 7.61–7.32 (m, 4H), 7.24–7.11 (m, 1H), 6.76 (d, *J* = 2.3 Hz, 1H), 6.73 (d, *J* = 8.9 Hz, 1H), 5.90 (br s, 2H), 1.30 (s, 9H); ^13^C NMR (63 MHz, CDCl_3_) δ 149.3, 135.5, 133.9, 132.4, 131.0, 130.9, 130.8, 129.9, 129.7, 127.1, 120.6, 118.6, 117.4, 55.4, 22.4; elemental analysis (%) calculated for C_17_H_18_C_l2_N_2_OS: C 55.29, H 4.91, N 7.59, S 8.68; found: C 55.06, H 4.97, N 7.71, S 8.82.

##### (*S*,*Z*)-*N*-((2-Amino-5-chlorophenyl)(2-chlorophenyl)methylene)-2-methylpropane-2-sulfinamide (***S*-1**)

[α]${}_{D}^{25}$ = +43.00 (c = 1.09 mg/mL, CHCl_3_); ^1^H NMR (250 MHz, CDCl_3_) δ 7.61–7.32 (m, 4H), 7.24–7.11 (m, 1H), 6.76 (d, *J* = 2.3 Hz, 1H), 6.73 (d, *J* = 8.9 Hz, 1H), 5.90 (br s, 2H), 1.30 (s, 9H).

#### 2.1.3. Diastereoselective Reduction of Sulfinylimine **2**

Sulfinylimine **1** (4.33 mmol) was dissolved in anhydrous THF (10 mL/mmol of **1**) under an argon atmosphere and cooled in ice bath. Lithium tri-*tert*-butoxyaluminum hydride (1M THF solution, 4.0 eq) was added dropwise, and the resulting solution was stirred and warmed to room temperature. After verifying imine consumption by TLC, the reaction was diluted with EtOAc (60 mL). The mixture was cooled to 0 °C, and water (20 mL) was added; then the organic phase was washed with brine, dried over anhydrous Na_2_SO_4_ and concentrated to dryness to give 692 mg (96%) of compound **2** as a single diastereomer, without the need for further purification.

##### (*R*)-*N*-[(*S*)-(2-Amino-5-chlorophenyl)(2-chlorophenyl)methyl]-2-methylpropane-2-sulfinamide (***R,S*-2**)

mp: 165 °C; IR (neat, cm^−1^): 3358, 3290, 1632, 1486, 1054; ^1^H NMR (250 MHz, CDCl_3_) δ 7.72 (dd, *J* = 7.9, 1.7 Hz, 1H), 7.43–7.27 (m, 3H), 7.06 (dd, *J* = 8.5, 2.4 Hz, 1H), 6.68 (d, *J* = 8.5 Hz, 1H), 6.46 (d, *J* = 2.4 Hz, 1H), 5.88 (d, *J* = 2.1 Hz, 1H), 4.28 (br s, 2H), 3.53 (d, *J* = 2.1 Hz, 1H), 1.27 (s, 9H); ^13^C NMR (63 MHz, CDCl_3_) δ 143.4, 137.8, 133.7, 130.2, 129.4, 129.4, 129.2, 127.8, 126.9, 125.7, 122.4, 118.1, 56.1, 54.2, 22.7; elemental analysis (%) calculated for C_17_H_20_Cl_2_N_2_OS: C 54.99, H 5.43, N 7.54, S 8.63; found: C 55.57, H 5.67, N 7.30, S 8.77.

##### (*S*)-*N*-[(*R*)-(2-Amino-5-chlorophenyl)(2-chlorophenyl)methyl]-2-methylpropane-2-sulfinamide(***S*,*R*-2**)

^1^H NMR (250 MHz, CDCl_3_) δ 7.80–7.63 (m, 1H), 7.43–7.21 (m, 1H), 7.01 (dd, *J* = 8.5, 2.4 Hz, 1H), 6.65 (d, *J* = 8.5 Hz, 1H), 6.45 (d, *J* = 2.4 Hz, 1H), 5.87 (d, *J* = 2.4 Hz, 1H), 4.60 (br s, 2H), 3.64 (d, *J* = 2.5 Hz, 1H), 1.25 (s, 9H).

#### 2.1.4. Cleavage of the *N*-*Tert*-Butylsulfinyl Chiral Auxiliary **3**

To a 1:1 mixture of 1,4-dioxane and MeOH (20 mL), cooled to 0 °C, we added acetyl chloride (4.0 eq.). After 15 min, compound **2** (2.69 mmol) was added, and the mixture was stirred at room temperature for 3 h. Then, the solvent was removed to dryness, and the residue was dissolved in water (20 mL), basified to pH 8 with concentrated aqueous Na_2_CO_3_ solution and extracted with EtOAc (3 × 40 mL). The combined organic layers were dried over anhydrous Na_2_SO_4_, and the solvent was removed under reduced pressure to afford 610 mg (85%) of the pure diamine **3** as a colorless oil.

##### (*S*)-2-[Amino(2-chlorophenyl)methyl]-4-chloroaniline (***S*-3**)

[α]${}_{D}^{25}$ = +36.00 (c = 1.00 mg/mL, CHCl_3_); IR (neat, cm^−1^): 3302, 2970, 1484, 1033; ^1^H NMR (250 MHz, CDCl_3_) δ 7.39 (app dt, *J* = 7.1, 2.4 Hz, 2H), 7.26 (m, 2H), 7.04 (dd, *J* = 8.4, 2.4 Hz, 1H), 6.94 (d, *J* = 2.4 Hz, 1H), 6.59 (d, *J* = 8.4 Hz, 1H), 5.50 (s, 1H), 3.01 (br s, 4H); ^13^C NMR (63 MHz, CDCl_3_) δ 144.1, 140.6, 133.5, 129.9, 128.9, 128.7, 128.5, 128.0, 127.6, 127.4, 122.8, 117.5, 53.5; elemental analysis (%) calculated for C_13_H_12_Cl_2_N_2_: C 58.45, H 4.53, N 10.49; found: C 58.28, H 4.39, N 10.79.

##### (*R*)-2-(Amino(2-chlorophenyl)methyl)-4-chloroaniline (***R*-3**)

[α]${}_{D}^{25}$ = −36.00 (c = 1.00 mg/mL, CHCl_3_) ^1^H NMR (250 MHz, CDCl_3_) δ; 7.40 (app dt, *J* = 7.1, 2.4 Hz, 2H), 7.27 (pd, *J* = 7.3, 3.5 Hz, 2H), 7.04 (dd, *J* = 8.4, 2.4 Hz, 1H), 6.94 (d, *J* = 2.4 Hz, 1H), 6.59 (d, *J* = 8.4 Hz, 1H), 5.50 (s, 1H), 4.37 (br s, 2H).

#### 2.1.5. Synthesis of Aminoethanol **4**

Compound **3** (490 mg, 1.8 mmol) was dissolved in dioxane (1 mL) in a microwave tube for 1 h under an argon atmosphere. Afterward, a 2.5–3.3 M ethylene oxide solution in THF (2.5 mL) and indium trichloride (0.18 mmol) were added. The mixture was irradiated at 100 °C. Then, the reaction was cooled to room temperature, diluted with ethyl acetate (30 mL) and washed with saturated sodium carbonate solution (10 mL) and water (10 mL). The organic layer was dried over anhydrous Na_2_SO_4_, and the solvent was removed under reduced pressure. The residue was purified by column chromatography on silica gel, using as eluent a 6:4 n-hexane/ethyl acetate mixture, to afford 220 mg (39%) of the desired compound.

##### (*S*)-2-(((2-Amino-5-chlorophenyl)(2-chlorophenyl)methyl)amino)ethan-1-ol (***S*-4**)

mp: 95 °C [α]${}_{D}^{25}$ = +81.00 (c = 0.9 mg/mL, CHCl_3_). Enantiomeric excess *ee* 93%, which was determined by chiral HPLC, buffer NaH_2_PO_4_ 20 mM pH 5.9/CH_3_CN 80:20, flow 1.0 mL/min, retention time 4.56 min (major), 5.21 min (minor); IR (neat, cm^−1^): 3234, 2925, 2848, 1485, 1468, 1034; ^1^H NMR (250 MHz, CDCl_3_) δ 7.44–7.33 (m, 2H), 7.32–7.18 (m, 2H), 7.03 (dd, *J* = 8.4, 2.5 Hz, 1H), 6.97 (d, *J* = 2.4 Hz, 1H), 6.60 (d, *J* = 8.4 Hz, 1H), 5.25 (s, 1H), 3.72 (t, *J* = 5.1 Hz, 2H), 2.90–2.65 (m, 2H). ^13^C NMR (63 MHz, CDCl_3_) δ 144.5, 138.1, 134.0, 130.1, 129.4, 129.0, 128.6, 128.2, 127.5, 127.0, 123.0, 117.8, 61.9, 60.6, 49.8; elemental analysis (%) calculated for C_15_H_16_Cl_2_N_2_O: C 57.89, H 5.18, N 9.00; found: C 55.10, H 4.91, N 8.35.

##### (*R*)-2-(((2-amino-5-chlorophenyl)(2-chlorophenyl)methyl)amino)ethan-1-ol (***R*-4**)

[α]${}_{D}^{25}$ = −78.00 (c = 1.00 mg/mL, CHCl_3_). Enantiomeric excess *ee* 95%, which was determined by chiral HPLC, buffer HNaPO_4_ 20 mM pH 5.9/CH_3_CN 80:20, flow 1.0 mL/min, retention time 4.60 min (minor), 5.02 min (major); ^1^H NMR (250 MHz, CDCl_3_) δ 7.44–7.33 (m, 2H), 7.31–7.18 (m, 2H), 7.03 (dd, *J* = 8.4, 2.5 Hz, 1H), 6.96 (d, *J* = 2.5 Hz, 1H), 6.59 (d, *J* = 8.4 Hz, 1H), 5.25 (s, 1H), 3.74–3.70 (dd, *J* = 4.8, 1.0 Hz, 1H), 3.69 (d, *J* = 4.8 Hz, 1H), 2.88–2.64 (m, 2H).

#### 2.1.6. Synthesis of Benzodiazepine **5**

To a stirred solution of compound **4** (420 mg, 1.35 mmol) in CH_2_Cl_2_ (15 mL) at 0 °C was added bromoacetyl bromide (305 mL, 3.5 mmol). Stirring was continued at 0 °C for 1 h and then at room temperature for 16 h. DIEA (3.5 mL, 20 mmol) was added, and the reaction was stirred at 50 °C for 5 h. The mixture was cooled to room temperature, diluted with CH_2_Cl_2_ (30 mL) and washed with water (20 mL). The organic layer was dried over anhydrous Na_2_SO_4_ and concentrated to dryness. The crude product was purified by using column chromatography on silica gel eluting with a gradient from hexane to 6:4 hexane/ethyl acetate to give 1.9 g (59%) of compound **5** as a white solid.

##### (*S*)-2-(7-Chloro-5-(2-chlorophenyl)-2-oxo-1,2,3,5-tetrahydro-4*H*-benzo[*e*][1,4]diazepin-4-yl)ethyl 2-bromoacetate (***S*-5**)

mp: 148–149 °C; IR (neat, cm^−1^): 2956, 1735, 1666, 905; ^1^H NMR (250 MHz, CDCl_3_) δ 8.49 (br s, 1H), 7.68–7.61 (m, 1H), 7.50–7.33 (m, 3H), 7.26 (dd, *J* = 8.5, 2.2 Hz, 1H), 7.02 (d, *J* = 8.5 Hz, 1H), 6.63 (d, *J* = 2.2 Hz, 1H), 5.31 (s, 1H), 4.43 (ddd, *J* = 11.7, 7.9, 4.3 Hz, 1H), 4.24 (dt, *J* = 11.7, 4.9 Hz, 1H), 3.88 (s, 2H), 3.58 (d, *J* = 15.5 Hz, 1H), 3.48 (dd, *J* = 15.5, 1.3 Hz, 1H), 3.09 (dt, *J* = 14.2, 4.3 Hz, 1H), 2.92 (ddd, *J* = 14.2, 7.9, 4.9 Hz, 1H). ^13^C NMR (63 MHz, CDCl_3_) δ 171.4, 167.2, 136.9, 136.6, 134.6, 132.7, 130.9, 130.8, 130.4, 130.3, 129.7, 129.1, 127.3, 122.2, 65.0, 63.9, 52.8, 52.0, 25.9; elemental analysis (%) calculated for C_19_H_17_BrCl_2_N_2_O_3_: C 48.33, H 3.63, N 5.93; found: C 48.18, H 3.52, N 5.82.

##### (*R*)-2-(7-Chloro-5-(2-chlorophenyl)-2-oxo-1,2,3,5-tetrahydro-4*H*-benzo[*e*][1,4]diazepin-4-yl)ethyl 2-bromoacetate (***R*-5**)

^1^H NMR (250 MHz, CDCl_3_) δ 8.90 (s, 1H), 7.66–7.59 (m, 1H), 7.47–7.33 (m, 3H), 7.26 (d, *J* = 2.3 Hz, 1H), 7.05 (d, *J* = 8.5 Hz, 1H), 6.64 (d, *J* = 2.2 Hz, 1H), 5.32 (s, 1H), 4.44 (ddd, *J* = 11.8, 7.9, 4.4 Hz, 1H), 4.24 (dt, *J* = 11.8, 4.9 Hz, 1H), 3.88 (s, 2H), 3.59 (d, *J* = 15.5 Hz, 1H), 3.49 (dd, *J* = 15.5, 1.1 Hz, 1H), 3.09 (dt, *J* = 14.2, 4.4 Hz, 1H), 3.02–2.86 (ddd, *J* = 14.2, 7.9, 4.9 Hz, 1H).

#### 2.1.7. Ester Hydrolysis **6**

Compound **5** (1 mmol) was dissolved in a 10% KOH methanol solution (10 mL). The reaction was stirred at room temperature while monitored by TLC, and additional 10% KOH methanol solution was periodically added until the starting material was no longer detected (4 h). Then, the reaction was diluted with water (20 mL), neutralized with 2M HCl solution and extracted with CH_2_Cl_2_ (3 × 40 mL). The combined organic layers were dried over sodium sulfate and filtered and the solvent was removed under vacuum. No further purification was needed.

##### (5*S*)-7-Chloro-5-(2-chlorophenyl)-4-(2-hydroxyethyl)-4,5-dihydro-1*H*-benzo[*e*][1,4]diazepin-2(3*H*)-one (***S*-6**)

mp: 163 °C; [α]${}_{D}^{25}$ = +114 (c = 1.00 mg/mL, CHCl_3_); Enantiomeric excess *ee* 93%, which was determined by chiral HPLC, buffer NaH_2_PO_4_ 20 mM pH 5.9/*i*-PrOH 80:20, flow 1.0 mL/min, retention time 4.643 min (minor), 5.090 min (major); IR (neat, cm^−1^): 3396, 2925, 1676, 1040; ^1^H NMR (250 MHz, CDCl_3_) δ 8.48 (s, 1H), 7.48–7.42 (m, 1H), 7.35–7.19 (m, 5H), 6.98 (d, *J* = 8.5 Hz, 1H), 6.82 (d, *J* = 2.3 Hz, 1H), 5.39 (s, 1H), 3.85–3.57 (m, 2H), 3.52 (s, 2H), 3.02–2.80 (m, 2H), 2.50 (br s, 1H); ^13^C NMR (63 MHz, CDCl_3_) δ 173.0, 137.0, 135.8, 134.4, 131.5, 131.1, 130.8, 130.7, 130.1, 129.8, 129.2, 127.4, 121.9, 65.8, 58.8, 55.1, 52.4.; elemental analysis (%) calculated for C_17_H_16_Cl_2_N_2_O_2_: C 58.14, H 4.59, N 7.98; found: C 58.46, H 4.72, N 7.74.

##### (5*R*)-7-Chloro-5-(2-chlorophenyl)-4-(2-hydroxyethyl)-4,5-dihydro-1*H*-benzo[*e*][1,4]diazepin-2(3*H*)-one (***R*-6**)

[α]${}_{D}^{25}$ = −116.00 (c = 1.04 mg/mL, CHCl_3_); Enantiomeric excess *ee* 97%, which was determined by chiral HPLC, buffer NaH_2_PO_4_ 20 mM pH 5.9/*i*-PrOH 80:20, flow 1.0 mL/min, retention time 4.427 min (major), 5.457 min (minor); ^1^H NMR (250 MHz, CDCl_3_) δ 8.48 (s, 1H), 7.48–7.42 (m, 1H), 7.35–7.19 (m, 5H), 6.98 (d, *J* = 8.5 Hz, 1H), 6.82 (d, *J* = 2.3 Hz, 1H), 5.39 (s, 1H), 3.85–3.57 (m, 2H), 3.52 (s, 2H), 3.02–2.80 (m, 2H), 2.50 (br s, 1H).

#### 2.1.8. Synthesis of Aza-CGP37157–Lipoic Hybrids (**7**)

The suitable compound **6** (50 mg, 0.14 mmol), α-lipoic acid (*R*)-(+)- enantiomer or (*S*)-(−)- enantiomer) (0.17 mmol) and DMAP (0.43 mmol) were dissolved in anhydrous dichloromethane (5 mL) in argon atmosphere. The mixture was cooled at 0 °C, and a solution of EDCI (0.17 mmol) in anhydrous dichloromethane (2 mL) was added dropwise. The reaction was stirred at room temperature for 16 h. Then, it was diluted with dichloromethane (30 mL), washed with water (3 × 10 mL), dried over anhydrous Na_2_SO_4_ and the solvent was removed under reduced pressure. The residue was purified by column chromatography on silica gel eluting with a gradient from hexane to 6:4 hexane/ethyl acetate to give 49 mg (65%) of compound **7** as a colorless viscous oil.

##### 2-((*S*)-7-Chloro-5-(2-chlorophenyl)-2-oxo-1,2,3,5-tetrahydro-4*H*-benzo[*e*][1,4]diazepin-4-yl)ethyl 5-((*S*)-1,2-dithiolan-3-yl)pentanoate (**7a**)

[α]${}_{D}^{25}$ = +165.00 (c = 1.00 mg/mL, CHCl_3_); IR (neat, cm^−1^): 2927, 1728, 1665, 1483, 1034; ^1^H NMR (300 MHz, CDCl_3_) δ 8.71 (s, 1H), 7.62–7.52 (m, 1H), 7.44–7.38 (m, 1H), 7.35–7.29 (m, 2H), 7.24 (dd, *J* = 8.5, 2.2 Hz, 1H), 7.00 (d, *J* = 8.5 Hz, 1H), 6.61 (d, *J* = 2.2 Hz, 1H), 5.30 (s, 1H), 4.34–4.23 (m, 1H), 4.20–4.05 (m, 1H), 3.61–3.42 (m, 4H), 3.22–2.97 (m, 3H), 2.95–2.79 (m, 1H), 2.53–2.38 (m, 1H), 2.32 (t, *J* = 7.3 Hz, 2H), 1.97–1.80 (m, 1H), 1.74–1.58 (m, 4H), 1.52–1.39 (m, 2H); ^13^C NMR (63 MHz, CDCl_3_) δ 173.4, 172.0, 137.0, 136.6, 134.6, 132.6, 130.8, 130.6, 130.4, 130.2, 129.6, 129.0, 127.1, 122.2, 65.1, 62.1, 56.4, 53.0, 52.2, 40.3, 38.6, 34.7, 34.1, 28.9, 24.7; elemental analysis (%) calculated for C_25_H_28_Cl_2_N_2_O_3_S_2_: C 55.65, H 5.23, N 5.19, S 11.88; found: C 55.47, H 5.08, N 5.16, S 11.75. HRMS (MALDI-TOF): calcd. for C_25_H_27_Cl_2_N_2_O_3_S_2_ (M^+^), 537.0846; found, 537.0817.

##### 2-((*R*)-7-Chloro-5-(2-chlorophenyl)-2-oxo-1,2,3,5-tetrahydro-4*H*-benzo[*e*][1,4]diazepin-4-yl)ethyl 5-((*S*)-1,2-dithiolan-3-yl)pentanoate (**7b**)

[α]${}_{D}^{25}$ = +140.00 (c = 1.00 mg/mL, CHCl_3_); ^1^H NMR (300 MHz, CDCl_3_) δ 8.30 (br s, 1H), 7.63–7.54 (m, 1H), 7.44–7.39 (m, 1H), 7.38–7.27 (m, 2H), 7.24 (dd, *J* = 8.4, 2.2 Hz, 1H), 6.97 (d, *J* = 8.4 Hz, 1H), 6.61 (d, *J* = 2.2 Hz, 1H), 5.30 (s, 1H), 4.35–4.21 (m, 1H), 4.21–4.09 (m, 1H), 3.61–3.39 (m, 3H), 3.22–2.96 (m, 3H), 2.94–2.78 (m, 1H), 2.51–2.39 (m, 1H), 2.32 (td, *J* = 7.3, 1.6 Hz, 2H), 1.96–1.83 (m, 1H), 1.72–1.58 (m, 4H), 1.52–1.38 (m, 2H). HRMS (MALDI-TOF): calculated for C_25_H_27_Cl_2_N_2_O_3_S_2_ (M^+^), 537.0846; found, 537.0815.

##### 2-((*S*)-7-Chloro-5-(2-chlorophenyl)-2-oxo-1,2,3,5-tetrahydro-4*H*-benzo[*e*][1,4]diazepin-4-yl)ethyl 5-((*R*)-1,2-dithiolan-3-yl)pentanoate (**7c**)

[α]${}_{D}^{25}$= −138.00 (c = 0.9 mg/mL, CHCl_3_) ^1^H NMR (300 MHz, CDCl_3_) δ 8.31 (br s, 1H), 7.56–7.47 (m, 1H), 7.38–7.31 (m, 1H), 7.31–7.22 (m, 2H), 7.17 (dd, *J* = 8.5, 2.2 Hz, 1H), 6.91 (d, *J* = 8.5 Hz, 1H), 6.55 (d, *J* = 2.2 Hz, 1H), 5.23 (s, 1H), 4.30–4.14 (m, 1H), 4.14–4.00 (m, 1H), 3.55–3.32 (m, 3H), 3.16–2.88 (m, 3H), 2.87–2.67 (m, 1H), 2.47–2.31 (m, 1H), 2.25 (t, *J* = 7.3 Hz, 2H), 1.93–1.74 (m, 1H), 1.66–1.53 (m, 4H), 1.48–1.29 (m, 2H). HRMS HRMS (MALDI-TOF): calculated for C_25_H_27_Cl_2_N_2_O_3_S_2_ (M^+^), 537.0846; found, 537.0827.

##### 2-((*R*)-7-Chloro-5-(2-chlorophenyl)-2-oxo-1,2,3,5-tetrahydro-4*H*-benzo[*e*][1,4]diazepin-4-yl)ethyl 5-((*R*)-1,2-dithiolan-3-yl)pentanoate (**7d**)

[α]${}_{D}^{25}$ = −160.00 (c = 1.00 mg/mL, CHCl_3_); ^1^H NMR (300 MHz, CDCl_3_) δ 8.41 (br s, 1H), 7.62–7.51 (m, 1H), 7.45–7.38 (m, 1H), 7.37–7.27 (m, 2H), 7.24 (dd, *J* = 8.5, 2.2 Hz, 1H), 6.98 (d, *J* = 8.5 Hz, 1H), 6.61 (d, *J* = 2.2 Hz, 1H), 5.30 (s, 1H), 4.35–4.22 (m, 1H), 4.21–4.09 (m, 1H), 3.61–3.40 (m, 3H), 3.23–2.96 (m, 3H), 2.93–2.78 (m, 1H), 2.52–2.38 (m, 1H), 2.32 (td, *J* = 7.3, 1.5 Hz, 2H), 1.97–1.82 (m, 1H), 1.73–1.57 (m, 4H), 1.54–1.38 (m, 2H). HRMS (MALDI-TOF): calculated for C_25_H_27_Cl_2_N_2_O_3_S_2_ (M^+^), 537.0846; found, 537.0815.

### 2.2. Pharmacological Evaluation

#### 2.2.1. Physicochemical and ADME Properties Calculation

Physicochemical and ADME (absorption, distribution, metabolism and excretion) characteristics of compounds were calculated with SwissADME web server [21] and QikProp module of Schrodinger software [22]. Compounds were firstly prepared with LigPrep module at Maestro [22,23]. Protonation and tautomerization states were analyzed at pH 7.4, and then consequent 3D structures were minimized by using OPLS3 force field [24]. Finally, minimized 3D structures were evaluated for ADME prediction, using QikProp module of Schrodinger software.

#### 2.2.2. Reduction Assay of 1,1-Diphenyl-2-picryl-hydrazyl (DPPH): Antioxidant Capacity

Antioxidant capacity was evaluated by using described procedure slightly modified [25]. Compounds (0.1 and 1 mM, final concentration) were diluted in assay media (EtOH/H_2_O, 70:30) and mixed with 100 μL of DPPH (SigmaAldrich, D9132, Madrid, Spain) solution (100 μM final concentration) in a clear 96-multiwell plate. Final solution was kept in dark for 30 min, and absorbance of DPPH was measured in a SpectroStar Nano plate-reader (BMG Labtech, Ortenberg, Germany) in duplicate at 490 nM. All experiments included a blank (assay media), control (DPPH 100 μM) and a positive control (ascorbic acid, 100 μM). Data were evaluated as % absorbance (Abs) reduction of control after subtracting blank absorbance.
% DPPH reduction = [100 − ((Abss_ample_ − Abs_blank_) × 100]/Abs_control_

#### 2.2.3. Determination of NRF2 Transcription Factor Induction

AREc32 cell line was kindly shared by Professor Roland Wolf (University of Dundee, UK) [26]. AREc32 cells were maintained in DMEM/glutamax/high glucose (Gibco, 61965-06, Madrid, Spain), supplemented with 1% penicillin–streptomycin (10,000 units) (SigmaAldrich, P4333, Madrid, Spain), geneticin (0.8 mg/mL) (Gibco, 11811-031) and 10% FBS (Gibco, 10270106), at 37 °C in a 5% CO_2_ air atmosphere. Cells were used from passage 4 to 18, and mycoplasma contamination was tested every 4 weeks. AREc32 cells were seeded in 96-well white plates (2 × 10^4^ cells/well) for 24 h. Thereafter, cells were treated with each compound (0.3, 3, 10 and 30 µM) in duplicate for 24 h. Non-treated cells were included as basal luciferase expression and tert-butylhydroquinone (TBHQ, 10 μM) as positive control in each plate. Luciferase expression was evaluated by using the Luciferase Assay System (Promega, E1483, Madrid, Spain) by luminescence in an Orion II microplate luminometer (Berthold, Germany). Luciferase activity increase was normalized to basal conditions considered as 1. CD values (concentration required to double the luciferase activity) were calculated from dose–response curves of luciferase fold induction vs. compound concentration, fitted by non-linear regression and data interpolated to value 2.

#### 2.2.4. Immunocytochemistry

AREc32 cells were plated in 24 multi-well plates on poly-*D*-Lysine-coated crystal slides (4 × 10^4^ cells/well). Cells were treated with compounds **7a** or **7b** (30 µM) for 2 h and fixed by treatment with 4% paraformaldehyde solution in PBS (10 min). After 3 washings with PBS every 5 min, cells were permeabilized with 0.1% Triton X-100 solution for five min, washed 3 times with PBS (5 min) and incubated overnight with primary antibody anti-NRF2 (1:50, A-10, sc-365949, Santa Cruz Biotechnology, Dallas, TX, USA). Thereafter, cells were washed with PBS (3×, 5 min) and incubated with secondary antibody (1:500, 1 h). Finally, nuclei were stained with 5 µg/mL PBS solution of Hoechst 33342 (Invitrogen, Madrid, Spain) during the second wash. Finally, slides were mounted on coverslips (glycerol–PBS (1:1 *v*/*v*)), and images were obtained in a confocal microscope (TCS SPE, Leica, Wetzlar, Germany).

#### 2.2.5. Western Blot Analysis

AREc32 cells were seeded in 6 multi-well plates (1 × 10^6^ cells/well) for 24 h and treated with compounds **7a** and **7b** (30 µM) and culture medium (basal) for 24 h. Thereafter, cells were collected and lysed in ice-cold lysis buffer (10% glycerol, 137 mM NaCl, 1% Nonidet P-40, 20 mM Tris HCl pH 7.5, 1 mM phenylmethylsulfonyl fluoride, 20 mM NaF, 1 mM sodium pyrophosphate, 1 μg/mL leupeptin and 1 mM Na_3_VO_4_). Then proteins (30 μg) were resolved by gel electrophoresis on sodium dodecyl sulfate–polyacrylamide (10% and 12%) and transferred to Immobilion-P membranes (MilliporeSigma, Madrid, Spain). The membranes were incubated with anti-GCLc (1:1000, Ab41463 (Abcam, Cambridge, MA, USA), anti-HO-1 (1:1000, ab68477, Abcam) or anti-Actin (1:100,000, A3854, Merck, Madrid, Spain). Peroxidase-conjugated secondary antibodies (1:10,000 and antirabbit: SC-2357, Santa Cruz Biotechnology, Dallas, TX, USA) were employed to detect the proteins by enhanced chemiluminescence. Band intensities corresponding to immunoblot detection of protein samples were quantitated with Fiji software [27].

#### 2.2.6. SH-SY5Y Neuroblastoma Cell Culture

SH-SY5Y cells (ECACC; 94030304, Porton Down, Salisbury, UK) were maintained with F12 (SigmaAldrich, N3520) and MEM (Gibco, 61100-087) 1:1 mixture, supplemented with non-essential amino-acids (0.5%), 0.5 mM sodium pyruvate (SigmaAldrich, P5280), 100 µg/mL streptomycin, 100 units/mL penicillin and 10% fetal bovine serum (FBS) at 37 °C in humidified atmosphere 5% CO_2_. Cells were used from passages 5 to 14, and mycoplasma contamination was tested every 4 weeks.

#### 2.2.7. Neuroprotection Assays in the SH-SY5Y Cell Line

Human neuroblastoma SH-SY5Y cells were plated in 96-well clear multi-well plates (6 × 10^4^ cells/well). The potential neuroprotective activity of compounds against OS was evaluated using the rotenone (30 μM) (SigmaAldrich, R8875) and oligomycin A (10 μM) (SigmaAldrich, 75351) (R/O) toxic mixture. Neuroprotection activity against Tau hyperphosphorylation was evaluated by using the okadaic acid (20 nM) model. Cells were pre-incubated with each compound (1 µM) for 24 h, and then treatments were removed and cells were incubated with compounds (1 µM) in the presence the corresponding toxic for additional 24 h. At the end of the experiments, viability was determined by the MTT method. Data were normalized from cell death induced by the toxic stimuli (considered as 100% cell death). Then, the protection percentages were calculated by subtracting from the death induced by toxic stimuli, the death in the presence of the compound.

#### 2.2.8. Cytotoxicity Assay in the SH-SY5Y Cell Line (CC_50_)

SH-SY5Y human neuroblastoma cells were seeded in 96-well clear multi-well plates (6 × 10^4^ cells/well). Cells were treated with increasing concentrations of compounds (10, 30 and 100 μM) for 24 h. Thereafter, viability was assessed by the MTT method. Cytotoxic concentration to observe 50% mortality (CC_50_) values for compounds were interpolated from dose–response curves represented as percentage of cell death vs. concentration of compound, fitted by non-linear regression and data interpolated to value 50.

#### 2.2.9. Viability Assessment by MTT Reduction

The tetrazolium ring of MTT can be cleaved by active dehydrogenases to produce a formazan precipitate. At the end of the experiments, MTT (SigmaAldrich, M2003) stock solution was added to each well to achieve a final concentration of 0.5 mg/mL. Cells were incubated for 2 h. Thereafter, culture media was removed, and DMSO was added to each well to dissolve the formazan precipitate. Finally, absorbance was measured in a microplate reader (SPECTROstar nano, BMG Labtech) at 570 nm. Absorbance obtained in basal conditions was considered as 100% cell viability.

#### 2.2.10. BV2 Cell Line Culture

BV2 mouse microglial cells (ATCC; CRL-2469) were cultured in RPMI (SigmaAldrich, R6504), with 100 µg/mL streptomycin, 100 units/mL penicillin and 10% FBS. Cells were maintained at 37 °C in a humidified atmosphere and 5% CO_2_. Cells were used from passage 4 to passage 16.

#### 2.2.11. Nitrite Production Reduction Assay

BV2 microglial cells were seeded at 2 × 10^4^ cells/well and treated with compounds at increasing concentrations during 24 h in complete culture media. Then, treatments were removed, and BV2 cells were co-incubated with compounds and LPS (100 ng/mL) (O127:B8, SigmaAldrich, L3129) for additional 18 h in culture media supplemented with 1% FBS. Each plate included non-treated cells as basal nitrite production. Thereafter, nitrite production was determined by using the modified Griess assay. In brief, samples (150 μL) were mixed with DAPSONE (75 μL) (SigmaAldrich, A74807) and NEDA (75 μL) (SigmaAldrich, 33461), and the mixture was incubated at room temperature for 5 min. Light absorption was measured at 550 nm in a microplate reader (SPECTROstar nano, BMG Labtech). Basal nitrite production was referred to as 100% of nitrite production, and all data were normalized to the basal condition. Concentration needed to reduce nitrite production to 50% was calculated from graphical representation of percentage of nitrite production vs. compound concentration, and data were fitted by non-linear regression and interpolated to 50%.

#### 2.2.12. Statistical Analysis

Values are expressed as mean ± SEM. IC_50_, and CC_50_ parameters were calculated from individual concentration–response curves by performing non-linear regression analysis, using GraphPad Prism software (San Diego, CA, USA). Results were analyzed by comparing experimental and control data, using one-way ANOVA, followed by Newman–Keuls post hoc test when three groups are implicated. Differences were considered to be statistically significant if *p* ≤ 0.05; *n* represents the number of different cultures used or number of assays performed.

## 3. Results

### 3.1. Chemistry

After some unsuccessful enantioselective approaches to obtain aza-CGP37157 as a single enantiomer, we decided to use a previous method to obtain diamines by a diastereoselective sulfinylimine reduction induced by Ellman’s sulfinamide as a chiral auxiliary [28]. Sulfinylimine **1** was prepared from the commercially available benzophenone that was previously employed for the racemic synthesis and commercial (*R*)-*tert*-butylsulfinamide in the presence of titanium tetraethoxide under microwave irradiation (Figure 2).

Compound **1** was isolated exclusively as a *Z* isomer stabilized by an intramolecular H-bond. After a sulfinylimine **1** reduction optimization with different reagents (Table 1), lithium tri(*tert*-butoxy)aluminium hydride was chosen and afforded **2** in a fully diastereoselective reduction and quantitative yield. The high diastereoselectivity ratio was explained by the formation of six-membered transition state between **1** and the aluminum hydride (Figure 3). The bulkier groups were placed in equatorial positions and allowing a fully diastereo-controlled intramolecular hydride transfer to the imine bond, leading to the *S* absolute configuration when the (*R*)-*tert*-butylsulfinamide was used as starting material.

The sulfinamide function was then cleaved by in situ–generated HCl to give the diamine **3** in 85% yield. For the introduction of the 2-hydroxyethyl chain, this compound was treated with a solution of ethylene oxide in THF, in the presence of indium trichloride as a Lewis acid and under microwave irradiation, affording **4** in a moderate 39% yield because of the unavoidable formation of a disubstituted product. The diazepinone ring was generated in 56% yield by one-pot acylation/intramolecular alkylation of compound **5** with bromoacetyl bromide in the presence of diethyisopropylamine.

Finally, the ester moiety was hydrolyzed, giving the corresponding alcohol **6** (Figure 2). A similar process using the enantiomer of Ellman’s auxiliary afforded ent-**6**. Both compounds were esterified with (*R*)- or (*S*)-lipoic acid via activation of the carboxy group in the presence of a diimide (EDCI) and 4-dimethylaminopyridine (DMAP), yielding the enantiopure hybrids **7a–d** sowing all four possible combinations of stereocenter configuration (Figure 4, Appendix A).

### 3.2. Pharmacological Characterization

#### 3.2.1. Computational Druggability Study of Compounds **7**

The predicted physicochemical and ADME properties of compounds **7a–d** are summarized in Table 2. Overall, all reported compounds exhibited a good ADME profile. The predicted human oral absorption value, based on a quantitative multiple linear regression model which considers a value >80% as indicative of high oral absorption, shows that compounds **7a–d** are expected to have good oral bioavailability.

Compounds’ structural similarity to pan-assay interference compounds, as well as potential toxicity, was assessed by means of the Free ADMET Filtering for Drugs (FAF-Drugs4) tool [29] and the “False positive Remover” server [30]. None of the reported compounds was identified as PAINS by either of these methods. Additionally, none of the compounds exhibited any toxicity issues according to the FAF-Drugs4 software. It exclusively identified a low-risk structural alert associated with the presence of the chlorine atoms. Thus, the in silico analysis shows that the reported compounds do not exhibit any structural toxicity-related alerts and that the reported activities are presumably not due to PAINS.

#### 3.2.2. Blood–Brain Barrier Permeability

The prediction of the blood–brain barrier permeability of the compounds was assessed via three different parameters, PPMDCK, CNS MPO.v2 and CNS. PPMDCK is a predictor of the apparent permeability in the MDCK cell line, considered a representative model of the BBB. Compounds with a value of PPMDCK > 500 are considered highly prone to cross the BBB, while compounds with PPMDCK < 25 are predicted to have a very low permeability. The results show that all reported compounds exceed or are close to a value of 500, thus indicating that they have a high chance of being BBB permeable (Table 3). However, the PPMDCK parameter does not take into account active transport, which plays a major role in BBB permeability. To overcome this liability, we additionally assessed compounds’ CNS accessibility by using the CNS multi-parameter optimization version 2 (CNS MPO.v2) algorithm. The CNS MPO algorithm was originally developed by Wager et al. [31,32] by using six critical physicochemical properties, namely MW, logP, logD, TPSA, H-bond donors and pK_a_. Each of these properties is transformed into a function that defines undesirable and desirable range of values, providing a score from 0 to 1, respectively.

The summation of each of these components leads to a single numerical value from 0 to 6 that relates to compounds’ CNS-drug likeness. Application of this algorithm to a set of CNS-marketed drugs and Pfizer CNS candidates showed that 74% of the CNS-marketed drugs and 60% of Pfizer CNS candidates had a CNS MPO value ≥ 4. Application to CNS MPO.v2 to reported compounds showed that compounds **6** exhibit a CNS MPO.v2 score ≥ 5, indicating that they have a high theoretical probability of reaching the CNS. As an example, 83% of compounds with CNS MPO.v2 value > 5 were brain penetrant in one of the datasets employed in CNS MPO.v2 development. On the other hand, compounds **7a–d** show only moderate CNS MPO.v2 values, given their higher molecular weight, lipophilicity and TPSA. Prediction of the CNS activity was additionally assessed by using QikProp CNS parameter, based on a modified version of the logBB values reported by Luco et al. [33] and Kelder et al. [34]. The results indicate that all the compounds are theoretically active in the CNS (CNS parameter = 1), and this is in good agreement with the aforementioned predictions. Overall, these results show that the reported compounds have an adequate ADME profile which provides them a high probability of crossing the BBB and efficiently accessing the CNS.

#### 3.2.3. Antioxidant Capacity of Aza-CGP37157-LA **7a–d** Derivatives

Initially, we evaluated the free-radical-scavenging capacity of derivatives **7a–d** in the DPPH assay, based on the capacity of the tested derivatives to react with the 2,2-diphenyl-1-picrylhydracil radical to generate a non-radical derivate [35]. Hybrid derivatives **7a–d** were evaluated at two different concentrations (0.1 and 1 mM), considering the activities of previously reported racemic derivatives. IC_50_ values are reported for highly active compounds. Ascorbic acid was used as a positive control, and CGP37157, ***rac-*****3a** and **(*****rac,R*)-3b** were included as references. As depicted in Table 4, compounds **6** and **7a–d** were moderate DPPH radical-derived scavengers that were able to trap up to 35.9% (**(*R,S*)-7c**) free radicals at the highest concentration. Those results are similar to those obtained for racemic mixtures **3a** and **3b** and a mixture of **(*S,R*)-7b** and **(*R,R*)-7d** obtained from ***rac***-**3** and *R*-lipoic acid.

### 3.3. Biological Evaluation

#### 3.3.1. Cytotoxicity Evaluation in the SH-SY5Y Cell Line

In order to evaluate the potential pharmacological profile of new compounds, we first tested the potential neurotoxicity in the neuroblastoma SHSY5Y cell line. Toxicity was evaluated at increasing concentrations (10, 30 and 100 μM), and toxicity was measured as MTT reduction capability. CC50 values were calculated by non-linear regression of curves % survival vs. concentration (Appendix A). CGP37157 and previously studied racemic derivatives ***rac-*****3a** and **(*****rac,R*)-3b** were included as reference compounds. As previously described, LA is a non-neurotoxic compound at the concentrations texted [36]. CGP37157 demonstrated a cytotoxic dose to induce a 50% viability reduction (CC_50_) of 57.3 ± 2.3 µM, which is in agreement with previously reported neurotoxicity [37]. Hybrid racemic derivatives showed an improved safety profile compared to parent compound CGP37157, showing CC_50_ values of 97.1 ± 2.3 µM and 76.2 ± 4.8 µM for ***rac-*****3a** and **(*****rac,R*)***-***3b** hybrid compounds, respectively (Table 5). Regarding enantiopure derivatives **7a–d**, all compounds showed an improved safety profile, being non-toxic at all concentrations tested (CC_50_ > 100 µM).

#### 3.3.2. NRF2 Induction

As previously reported, derivatives CGP37157, ***rac-*****3a** and **(*****rac,R*)-3b** demonstrated NRF2 induction activity at the micromolar range. More importantly, both compounds showed interesting differences depending on the configuration of chiral carbon at LA [19]. NRF2 has been associated with the LA moiety that has previously demonstrated a capacity to activate the phase II antioxidant response [38,39], but at high concentrations [19]. To evaluate the influence of both stereogenic centers in the NRF2 induction capacity of aza-CGP37157-LA hybrids, we used the AREc32 cell line model [26] by analyzing luciferase expression upon treatment with NRF2 inducers. AREc32 cells were exposed to increasing concentrations of tested compounds **7a–d** (0.3, 1, 10 and 30 μM) during 24 h; thereafter, luciferase expression was evaluated by luminescent assay to calculate the corresponding CD value (concentration to double luciferase expression). Used concentrations showed no toxicity or low toxicity in the AREc32 cell line (Appendix A). TBHQ was included as a positive control, and CGP37157, *S*- and *R*-LA, **rac-****3a** and **(*****rac,R*)-3b** and precursors derivatives (***R*-6** and ***S*-6**) were included for comparative purposes. Previous results demonstrated an improved NRF2 induction capacity when racemic LA was exchanged by the *R*-LA moiety (CD = 14.8 ± 1.6 μM and CD = 9.8 ± 1.6 μM, respectively [19]). In line with these results, enantiopure derivatives **7a–d** also showed interesting differences in activity depending on both chiral centers. As shown in Figure 5 and Appendix A, derivatives with the *S* configuration at the aza-CGP37157 analogue moiety showed improved potency compared to the corresponding *R* derivatives at the same position, with the latter being poorer NRF2 inducers with CD values over 30 μM, although they were able to significantly increase luciferase activity at 10 μM (**(*****R,S*)-7c**) and 30 μM (**(*R,R*)-****7d** and **(*****R,S*)-7c**). Interestingly, enantiopure aza-CGP37157 derivatives **(*****R*)-6** and **(*****S*)-6** showed improved NRF2 induction capacity compared to its racemic mixture **(±)-6** [19], demonstrating an important dependence of this pharmacological activity on the configuration of C1 at the aza-CGP37157 moiety. This result might be related to a mechanism of action that depends on the interactions with KEAP1 protein and deserves to be further elucidated. Compared to CGP37157, both compounds showed a highly improved NRF2 induction ability, as this compound showed a slight induction capacity at 10 μM that was drastically reduced at 30 μM [37]. Regarding **(*S,R*)-7b** and **(*S,S*)-7a** derivatives, bearing an *S*-aza-CGP moiety, both compounds showed concentration-dependent NRF2 induction capacity being derivative **(*S,R*)-7b** 2.2-fold more potent than **(*S,S*)-7a**. This result is in correlation with previously described activities for **rac-****3a** and **(*****rac,R*)-3b**, in which the compound bearing the *R*-LA moiety showed increased potency compared to the racemic mixture of diastereomers [19]. These results might be related with the metabolic processing of *R*- and *S*-LA [40,41,42] by mitochondrial lipoamide dehydrogenase that converts *R*-LA to dihydrolipolic acid 28-times faster than the *S*-LA derivative [43]. Once *R*-LA is reduced, it might activate the phase II antioxidant response by acting as pro-oxidant over key Cys residues at KEAP1 to generate the corresponding lipoyl-cystinyl disulphide [44]. All together, these results and their interesting structure–activity relationships demonstrate the dependence of their NRF2-induction capability on the configuration of chiral carbons included in their structure.

#### 3.3.3. Compounds **7a** and **7b** Induce NRF2 Nuclear Translocation and Upregulate the Expression of NRF2 Dependent Genes

To obtain deeper insight into the mechanism of action of compounds **7**, we evaluated their potential capacity to induce NRF2 nuclear localization and the activation of the phase II response. As previously described, NRF2 is sequestered at the cytoplasm by its repressor protein KEAP1, which mediates its proteasomal degradation. In the presence of high oxidative stress or NRF2 inducers, NRF2 is liberated and quickly translocates to the nucleus. Thus, we firstly evaluated the capacity of compounds **7a** and **7b** to induce NRF2 nuclear localization as part of their mechanism of action. AREc32 cells were treated with compounds **7a** and **7b** (30 μM) or culture media (Basal) for 2 h. Then the cells were fixed and processed for double staining with anti-NRF2 and Hoechst 33342. As observed in Figure 6A, both compounds induced a predominantly nuclear localization of NRF2 compared to basal conditions.

Once in the nucleus, NRF2 binds to small MafG protein, and the resulting complex binds to the ARE sequences at the DNA to initiate the expression of phase II antioxidant proteins. HO-1 and GCLc are two prominent examples of antioxidant enzymes regulated by the NRF2-ARE pathway related to the heme group processing and the synthesis of glutathione, the major antioxidant defense inside the cells. To demonstrate the capacity of compounds **7a** and **7b** to induce the expression of phase II antioxidant enzymes, AREc32 cells were treated with each compound at 30 μM or culture media (basal) for 24 h. Thereafter, protein expression was evaluated. As shown in Figure 6B,C, compounds **7a** and **7b** significantly increased the expression of HO-1 and GCLc in AREc32 after 24-h treatment (Appendix A). Therefore, these results demonstrate the ability of compounds **7a** and **7b** to activate the phase II antioxidant response.

#### 3.3.4. Anti-Inflammatory Properties of Compounds **7a–d**

Chronic neuroinflammation is considered a key pathological event in the early stages of AD development [45]. Among immune cells present at CNS, microglia is related to host defense and tissue repair [46]. Nonetheless, chronic neuroinflammation induced by an adverse environment of protein aggregates and exacerbated OS induces a neurotoxic activity of microglia that, finally, leads to neuronal death [45]. It is increasingly demonstrated that Aβ peptides are able to induce microglial proinflammatory phenotype by activation of different receptors, such as TLR4 [47], CD36 [48] and NLRP3 [49], fostering the release of pro-inflammatory cytokines and chemokines that further activate astrocytes generating a vicious cycle increasing neuronal death. Previously, we demonstrated the anti-inflammatory potential of the aza-CGP37157-LA hybrid derivatives [19], determining that racemic mixture **rac-****3a** was able to reduce nitrite production in primary glial cultures stimulated by LPS, while its diastereoselective derivative **(*****rac,R*)-3b** was not active. Intrigued by these results, we became interested in evaluating the anti-inflammatory properties of diastereoselective derivatives **7a–d** in order to elucidate the dependence of this pharmacological activity with the configuration of both stereogenic centers. To this end, we selected the BV2 microglial cell model stimulated by lipopolysaccharide (LPS), a TLR4 agonist that activates the pro-inflammatory response. CGP37157, *R-* and *S*-LA; analogue compounds ***rac-*****3a** and **(*****rac,R*)-3b**; and precursor derivatives **6** were included with comparative and reference purposes. Microglial BV2 cells were treated with increasing concentrations of the corresponding compound (0.3, 3, 10 and 30 µM) for 24 h following a pre-incubation protocol. Thereafter, treatments were exchanged by a combination of the corresponding compound at desired concentration and LPS (100 ng/mL) during 18 h. At the end of this period, nitrite production was evaluated by the Griess method as an indirect measurement of iNOS enzyme expression and BV2 activation [50]. In agreement with previous reported results [19], diastereoselective derivatives **7a–d** showed moderate anti-inflammatory properties. Interestingly, compound **(*S,R*)-7b** showed the highest anti-inflammatory capacity with an IC_50_ = 13.3 ± 3.68 μM, while diastereomers **(*S,S*)-7a**, **(*R,R*)-7d** and **(*R,S*)-7c** showed no significant activity at the evaluated concentrations (Table 6). These results are in line with NRF2 induction capacity in which derivative **(*S,R*)-7b** showed the best CD value (CD = 11.9 ± 1.3 μM).

#### 3.3.5. Neuroprotection in a Rotenone/Oligomycin A Oxidative Stress Model

As previously described, neuronal loss is the most important observation in AD associated to a complex network of pathological events in which OS, chronic neuroinflammation and aberrant protein aggregation play a key role in neuronal toxicity. Considering their antioxidant capacity, NRF2 induction properties and the anti-inflammatory capacity, we envisaged to evaluate the neuroprotective capacity of our enantiopure derivatives against OS. We used the rotenone/oligomycin A combination (R/O), a highly validated model of mitochondria intoxication to induce the overproduction of free radicals to simulate the exacerbated OS status widely described in AD patients [51,52]. These toxins inhibit mitochondrial complexes I and V, respectively, of the electron transport chain inducing the liberation of free radical species to promote OS and neuronal death.

Compounds **7a–d** have demonstrated moderate antioxidant capacity and diastereoselective NRF2 induction capacity. Thus, we selected a pre-incubation protocol to evaluate their potential neuroprotective capacity based on the NRF2-induction activity and direct antioxidant effect. Compound pre-incubation would induce the expression of antioxidant enzymes regulated by the phase II antioxidant response, thus promoting cells’ survival. The co-incubation period would also evaluate their capacity to trap free radicals produced by intoxicated mitochondria. SH-SY5Y neuroblastoma cells were treated with compounds (1 µM) during 24 h, and then cells were co-incubated with the corresponding compound at the same concentration and the R/O combination (30/10 µM) for additional 24 h. Melatonin, a well-known antioxidant natural neurohormone, was included as a positive control (1 µM). Previously reported racemic derivatives (**rac-****3a** and **(*****rac,R*)-3b**) and their precursors (**(****±****)-6**, ***R*****-6** and ***S*****-6**) were included for comparative purposes at the same concentration. The tested compounds showed, in general, interesting neuroprotective properties with protection percentages ranging from 44.1% (**(****±****)-6**) to 61.1% (**(*S,R*)-7b**) (Figure 7). In line with previous results [19], **rac-3a** and **(*rac,R*)-3b** showed similar neuroprotective capacity being compound derived from *R*-LA enantiomer slightly better neuroprotectant than ***rac*-3a**. This result is in line with the NRF2 induction capacity of both derivatives in which compound **(*****rac,R*)-3b**) demonstrated improved potency. Considering the configuration of stereocenters, the **(*S,R*)-7b** derivative showed the highest neuroprotection capacity, being able to improve cells’ survival by 61.1%. This result is in line with abovementioned higher potency to induce NRF2 compared to enantiopure diastereomers **(*S,S*)-7a**, **(*R,R*)-7d** and **(*R,S*)-7c** derivatives. These results indicate an interesting relationship of the pharmacological properties of compounds **7a–d** with their chirality.

#### 3.3.6. Neuroprotection against Tau Hyperphosphorylation Induced by Okadaic Acid

AD is characterized by the formation of neurofibrillary tangles that are composed of hyperphosphorylated Tau protein aggregates. Recent reports indicate that a high OS is related to the induction of Tau hyperphosphorylation linking protein aggregation, mitochondrial dysfunction and synaptic failure [53]. A direct correlation was found by inhibiting glutathione synthesis with buthionine sulfoximine, an in vitro model of OS, that greatly increased hyperphosphorylated Tau protein (^P^Tau) formation [54]. Moreover, the treatment of primary rat cortical neurons with cuprizone (cooper chelator) and OS induced by Fe^2+^/H_2_O_2_ combination significantly increased ^P^Tau formation via increased GSK3β activity, the principal kinase involved in Tau hyperphosphorylation [55]. In turn, ^P^Tau was demonstrated to facilitate copper reduction contributing drastically to OS by directly generating free radicals and initiating copper-canalized formation of H_2_O_2_. Furthermore, ^P^Tau activates p38 and RCAN1, the natural repressor of calcineurin, a phosphatase related to Tau dephosphorylation, accelerating the formation of ^P^Tau aggregates [56]. Therefore, OS-^P^Tau interconnection creates a pathological vicious cycle to induce neuronal degeneration in AD [57].

Considering these precedents, we envisaged the evaluation of our compounds in a hyperphosphorylation toxicity model induced by okadaic acid (OA), a protein phosphatase 2A inhibitor [58,59]. This inhibition causes an imbalance between kinase and phosphatase action, leading to hyperphosphorylation of proteins, in particular, Tau. Thus, treatment of SH-SY5Y neuroblastoma cells with OA is a well-stablished model of Tau hyperphosphorylation [60]. Similarly, to the method used for the OS model, we selected a pre-incubation/co-incubation protocol to evaluate the potential neuroprotection capacity related to the NRF2 induction/antioxidant properties of the compounds. Cells were treated with compounds **(****±****)-6**, ***R*****-6**, ***S*****-6** or **7a–d** (1 µM) during 24 h; thereafter, cells were co-treated with compounds (1 µM) and OA (20 nM) for a further 24 h. Melatonin was also included as a positive control. Finally, cellular viability was evaluated by the MTT method. As summarized in Figure 8, neuroprotection capacity against Tau hyperphosphorylation highly variated on the chiral configuration of the derivatives being **(*R,S*)-7c** derivative not protective while derivative **(*S,R*)-7b** showed a 41.9% protection. Interestingly, these results are also in agreement with aforementioned results in the OS model; **(*S,R*)-7b** shows the highest capacity to reduce cellular toxicity being the most potent NRF2 inducer. The neuroprotective activity was highly similar for derivative **(*R,S*)-7c**, although it was less potent NRF2 inducer, a result that indicates a potential secondary mechanism of action for these derivatives. In correlation with the loss of NRF2 induction potency, the **(*R,R*)-7d** derivative showed a 2-folds less potent capacity than the **(*S,R*)-7b** derivative, with a 21% protection. Finally, the **(*R,S*)-7c** diastereomer was not able to afford any protection, a result that might be in line with its low NRF2 induction capacity.

## 4. Conclusions

Considering the complex network of pathological events described for AD, we planned the development of a multitarget hybrid based on the pharmacological profile of CGP37157 and lipoic acid. Previously, we reported the synthesis and pharmacological characterization of the racemic mixture of an aza-analogue of CGP37157 hybridized with LA, including the racemic mixture and the natural *R*-enantiomer [19]. The results demonstrated the ability of these compounds to induce the activation of the phase II antioxidant response, antioxidant capacity and neuroprotective capacity. Interestingly, only the racemic derivative ***rac*-3a** afforded a moderate anti-inflammatory capacity, while derivative **(*rac,R*)-3b** bearing the *R*-LA moiety showed improved potency as a NRF2 inducer. Intrigued by these results, we studied each diastereomer in enantiopure form in order to evaluate the potential differences derived from the different configuration of both chiral centers. Interestingly, their pharmacological evaluation as NRF2 inducers revealed a strong dependence of this activity on the configuration of chiral centers. In this case, *S*-aza-CGP37157 analogs showed higher potency than their *R*-analogues, a highly interesting result that indicates a specific interaction in a highly restricted position. Considering the chiral center present at LA moiety, NRF2 induction capacity was also dependent on this position, reinforcing our previous conclusion. Hybrids **7a–d** were also moderate antioxidants that were able to reduce DPPH-derived free radicals with a similar potency to racemic derivatives, and, more importantly, compound **(*S,R*)-7b** showed a good anti-inflammatory profile that correlates with its higher capacity to induce the phase II antioxidant response. In contrast, *S*,*S*-, *R*,*S*- and *R*,*R*-derivatives show poorer anti-inflammatory properties. These results suggest a correlation between these two activities. Finally, our derivatives showed a good neuroprotective profile, being remarkable the neuroprotection afforded against exacerbated OS, in line with their antioxidant and NRF2 induction capacity. Compound **(*S,R*)-7b** showed, also, the best neuroprotective capacity. Considering the protection afforded against protein hyperphosphorylation, in general, these derivatives were poorer neuroprotectant agents; nonetheless, again, compound **(*S,R*)-7b** showed the best neuroprotective capacity. In conclusion, considering the interesting multitarget profile of compound **(*S,R*)-7b** and the dependence of biological activities with chiral centers configuration, we consider this compound to be a highly interesting hit, and its mechanism of action characterization will be evaluated and communicated in due course.

## Figures and Tables

**Figure 1 antioxidants-11-00112-f001:**
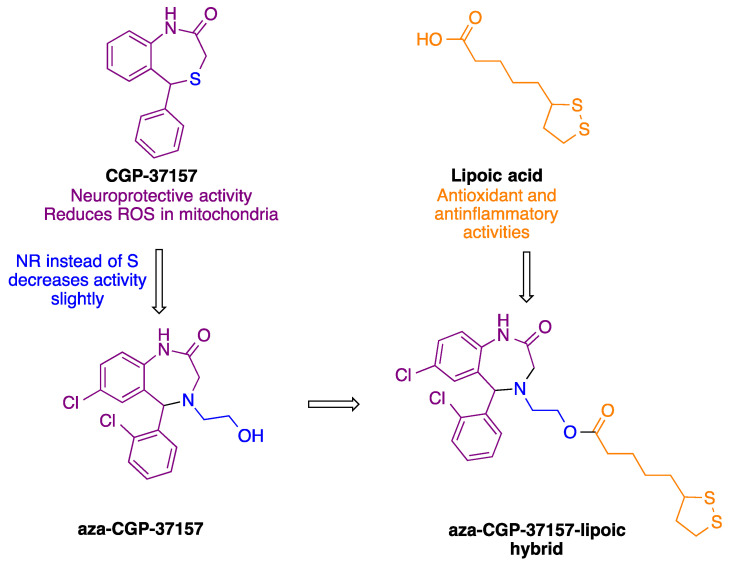
Design of the aza-CGP37157–lipoic acid hybrids described in this article.

**Figure 2 antioxidants-11-00112-f002:**
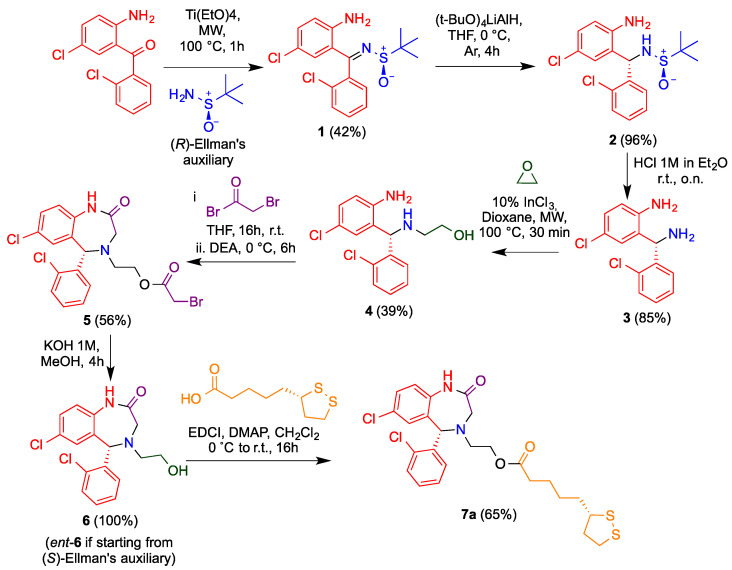
Synthetic pathway to obtain enantiopure aza-CGP37157–lipoic hybrids, as exemplified by the synthesis of **7a**. The route is based on the use of Ellman’s auxiliary to induce diastereoselectivity in the reduction of imine **1**.

**Figure 3 antioxidants-11-00112-f003:**
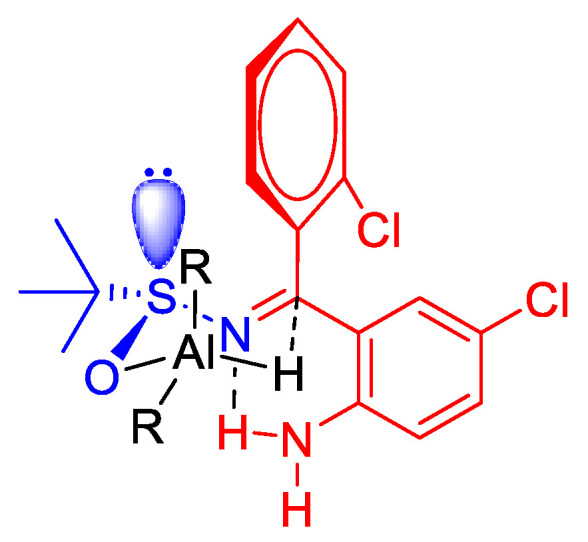
Transition state that explains the diastereoselective reduction of **1** under the influence of the stereocenter in Ellman’s auxiliary.

**Figure 4 antioxidants-11-00112-f004:**
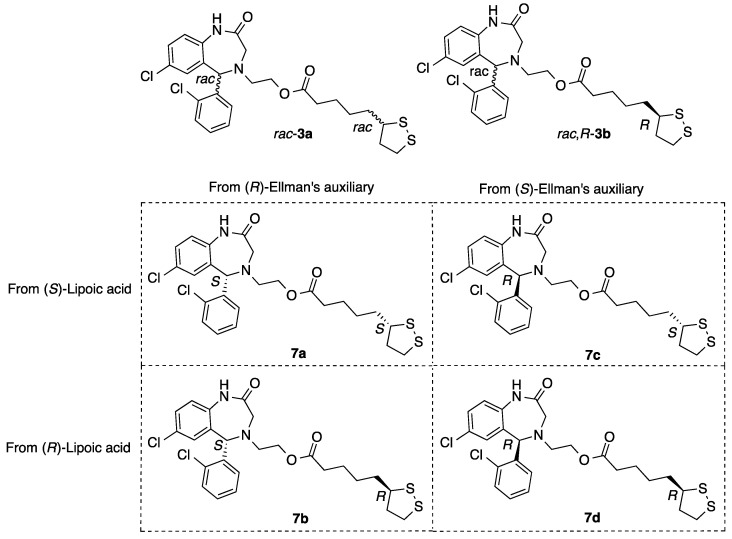
Structures of the four aza-CGP37157–lipoic hybrids studied in this work (**7a**–**7d**). Two previously studied, namely the fully racemic compound (**rac-3a**) and another one arising from a racemic diazepine and (*R*)-lipoic acid (**rac,*R*-3b**), were also studied as references.

**Figure 5 antioxidants-11-00112-f005:**
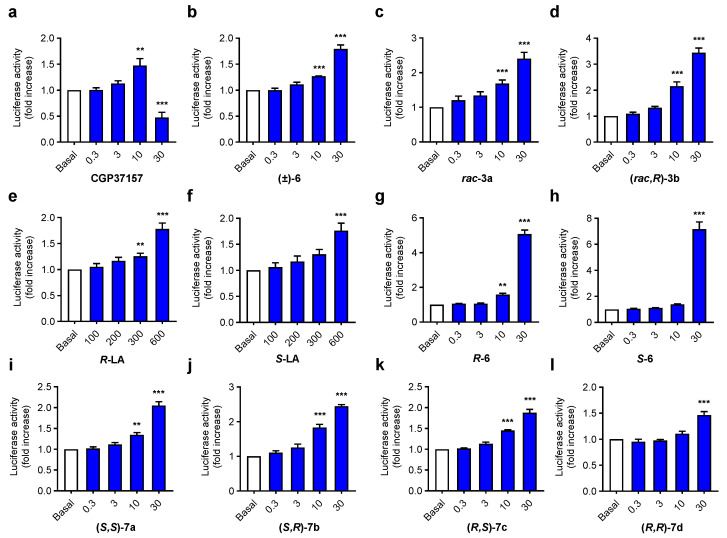
NRF2 induction capacity of compounds **7a–d** and their precursors **6**
*R*- and *S*-LA in the AREc32 cell line. (**a**–**l**) graph bar representation of NRF2 induction capacity of compounds **7a–d**, **6**, LA, **3a-b** and CGP37157, measured as luciferase expression in AREc32 cells and referenced to basal conditions AREc32 cells were treated with the corresponding compound at desired concentrations (0.3, 3, 10 and 30 µM), (LA derivatives: 0.1, 0.2, 0.3 and 0.6 mM) or culture medium (basal) for 24 h. Thereafter, luciferase expression was evaluated. Data are expressed as folds of increase normalized to basal conditions considered as 1. Data are means ± SEM of four independent experiments in duplicate; *** *p* < 0.001 and ** *p* < 0.01 compared to the corresponding basal conditions.

**Figure 6 antioxidants-11-00112-f006:**
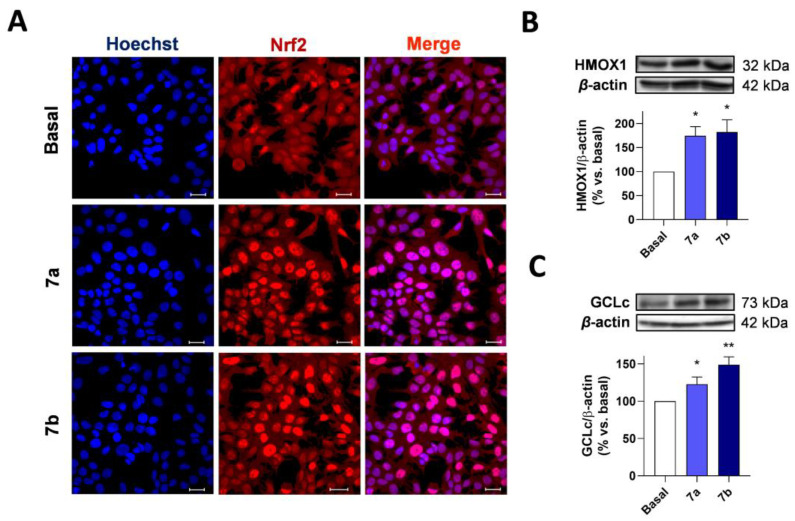
Compounds **7a** and **7b** induce NRF2 nuclear localization and overexpression of NRF2-regulated genes in AREc32 cells. (**A**) AREc32 cells were treated with compounds **7a** and **7b** (30 μM) or culture medium (basal) for 2 h and then processed for immunocytochemistry being stained with anti-NRF2 (red) and Hoechst (blue). HMOX1 (**B**) and GCLc (**C**) expression in AREc32 cells treated with culture medium (basal) or compounds **7a** and **7b** (30 μM) during 24 h. Graphs are represented as densitometric quantification, using β-actin for normalization. Scale bar: 25 μm. Data are expressed as mean ± SEM. of four independent experiments. Statistical differences were assessed by using one-way ANOVA: ** *p* < 0.01 and * *p* < 0.05 vs. basal condition.

**Figure 7 antioxidants-11-00112-f007:**
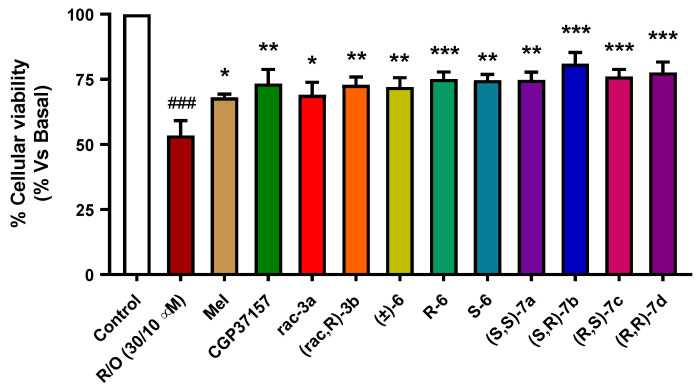
Neuroprotective effect of compounds **7a–d** against toxicity induced by high OS in the SH-SY5Y neuroblastoma cell line. SH-SY5Y cells were treated with the corresponding compound (1 µM) during 24 h. Then, cells were treated with the corresponding compound and the rotenone (30 µM) and oligomycin A (10 µM) combination. Melatonin was included as positive control, and compounds CGP37157, ***rac*-3a** and **(*rac,R*)-3b**, ***R*-6** and ***S*-6** were included as reference and comparative purposes. Data were obtained from five experiments by duplicate, with normalized vs. basal condition considered as 100% cellular viability and expressed as mean ± SEM. Statistical analysis was performed by using one-way ANOVA, followed by Newman–Keuls post-test: ^###^ *p* < 0.001 compared to basal; * *p* < 0.05, ** *p* < 0.01, *** *p* < 0.001 compared to toxic stimuli. Numerical data are given in Appendix A.

**Figure 8 antioxidants-11-00112-f008:**
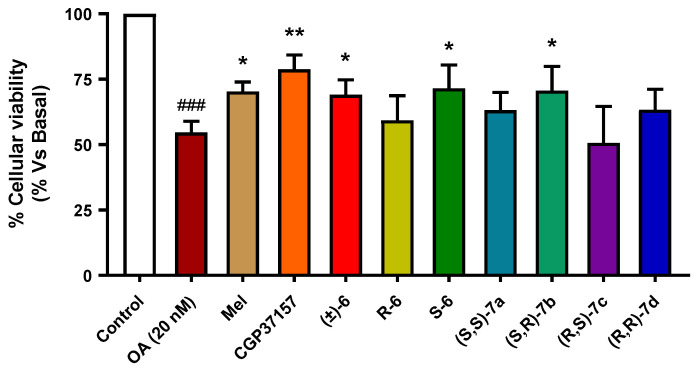
Neuroprotective effect of compounds **7a–d** against toxicity induced by protein hyperphosphorylation in the SH-SY5Y neuroblastoma cell line. SH-SY5Y cells were treated with the corresponding compound (1 µM) during 24 h. Then, cells were treated with a combination of the corresponding compound and okadaic acid (OA, 20 nM). Melatonin was included as control, and compounds CGP37157, ***R*-6** and ***S*-6** were included as reference and for comparative purposes. Data were obtained from five experiments by duplicate, with normalized vs. basal condition considered as 100% cellular viability, and expressed as mean ± SEM. Statistical analysis was performed by using one-way ANOVA, followed by Newman–Keuls post-test: ^###^ *p* < 0.001 compared to basal; * *p* < 0.05, ** *p* < 0.01 compared to toxic stimuli. Numerical data are given in Appendix A.

**Table 1 antioxidants-11-00112-t001:** Hydride diastereoselective and yield optimization.

Entry	Reagent	Yield (%)	dr ^1^
1	NaBH_4_	quant.	71:29
2	L-selectride	0	--
3	DIBAL	quant.	100:0
4	(*t*-BuO)_3_LiAlH	quant.	100:0

^1^ Calculated by ^1^H-NMR.

**Table 2 antioxidants-11-00112-t002:** Physicochemical and ADME properties of aza-CGP37157-LA **7a–d** derivatives.

Entry	Compound	MW (g/mol) ^a^	TPSA (Å^2^) ^a^	HBD ^a^	HBA ^a^	cLog P ^a^	Oral Absorption ^b^
1	**CGP37157**	324.22	54.40	1	1	3.9	100
2	***R*-6**	351.23	52.57	2	3	2.8	83
3	***S*-6**	351.23	52.57	2	3	2.8	85
4	**(*S,S*)-7a**	539.53	109.24	1	4	5.1	75
5	**(*S,R*)-7b**	539.53	109.24	1	4	5.2	75
6	**(*R,S*)-7c**	539.53	109.24	1	4	5.1	76
7	**(*R,R*)-7d**	539.53	109.24	1	4	5.2	81

MW = molecular weight, HBD = number of H-bond donors, HBA = number of H-bond acceptors, TPSA = topological polar surface area, cLog P = predicted octanol/water partition coefficient, oral absorption = predicted human oral absorption (0–100% scale). ^a^ Properties calculated with the SwissADME web server. ^b^ Properties calculated with the QikProp module of Schrodinger software.

**Table 3 antioxidants-11-00112-t003:** CNS-cross “in silico” prediction calculation.

Entry	Compound	PPMDCK (nm/s) ^a^	CNS MPO.v2	CNS Prediction ^b^
1	**CGP37157**	7461	5.2	CNS +
2	***R*-6**	465	5	CNS +
3	***S*-6**	668	5	CNS +
4	**(*S,S*)-7a**	1187	3	CNS +
5	**(*S,R*)-7b**	1203	3	CNS +
6	**(*R,S*)-7c**	1280	3	CNS +
7	**(*R,R*)-7d**	2613	3	CNS +

PPMDCK = predicted apparent MDCK cell line permeability. ^a^ Properties calculated with the QikProp module of Schrodinger software. ^b^ CNS activity prediction according to QikProp CNS parameter.

**Table 4 antioxidants-11-00112-t004:** Antioxidant activity of compounds toward DPPH-derived free radicals.

Entry	Compound	DPPH		
		Scavenging at 100 µM, %	Scavenging at 1mM, %	IC_50_, μM
1	**Trolox**	93.4 ± 0.5	92.2 ± 0.5	11.4 ± 1.0
2	**Ascorbic acid**	-	-	16.2 ± 0.7
3	**CGP37157**	9.6 ± 1.8	24.3 ± 1.4	-
4	***rac*-3a** [19]	15.1 ± 3.4 [19]	37.4 ± 3.8 [19]	-
5	**(*rac,R*)-3b** [19]	8.7 ± 5.1 [19]	30.7 ± 3.9 [19]	-
6	***R*-6**	10.30 ± 1.10	26.7 ± 1.3	-
7	***S*-6**	8.47 ± 2.42	25.3 ± 1.5	-
8	**(*S,S*)-7a**	10.9 ± 3.5	30.3 ± 1.6	-
9	**(*S,R*)-7b**	8.46 ± 3.8	32.0 ± 1.7	-
10	**(*R,S*)-7c**	13.6 ± 3.8	35.9 ± 1.9	-
11	**(*R,R*)-7d**	8.66 ± 3.0	23.3 ± 3.5	-

Data are expressed as means ± SEM of three different experiments in duplicate.

**Table 5 antioxidants-11-00112-t005:** Neurotoxicity of compounds in SH-SY5Y cell line. Cytotoxicity elicited by compounds in the neuroblastoma cell line SH-SY5Y measured as MTT reduction in presence of increasing concentrations of compound (10, 30 and 100 μM). Values are expressed as CC_50_ calculated from dose–response curves. Data are expressed as mean of three different experiments in duplicate.

Entry	Compound	CC_50_ (μM)
1	**CGP37157**	57.3 ± 2.3
2	***rac*-3a** [19]	97.1 ± 2.3
3	**(*rac,R*)***-***3b** [19]	76.2 ± 4.8
4	***R*-6**	>100
5	***S*-6**	>100
6	**(*S,S*)-7a**	>100
7	**(*S,R*)-7b**	>100
8	**(*R,S*)-7c**	>100
9	**(*R,R*)-7d**	>100

**Table 6 antioxidants-11-00112-t006:** Anti-inflammatory activity of compounds **7a–d** against LPS-induced activation of BV2 microglial cells.

Entry	Compound	IC_50_ (μM) BV2
1	**CGP37157**	21.9 ± 5.01
2	**(*S*)-Lipoic acid**	>30
3	**(*R*)** **-Lipoic acid**	>30
4	***rac*-3a**	13.3 ± 3.65
5	**(*rac,R*)-3b**	27.4 ± 1.60
6	***R*-6**	28.5 ± 1.21
7	***S*-6**	19.9 ± 5.66
8	**(*S,S*)-7a**	>30
9	**(*S,R*)-7b**	13.3 ± 3.68
10	**(*R,S*)-7c**	>30
11	**(*R,R*)-7d**	>30

IC_50_ values were obtained from nitrite production reduction elicited by compounds after LPS stimulation. BV2 microglial cells were pretreated with compounds (0.3, 3, 10 and 30 μM) for 24 h, and then the medium was replaced with fresh medium containing compounds and LPS (100 ng/mL) for 18 h. Nitrite production was assessed by the Griess method. Data are means of four different experiments in duplicate.

## Data Availability

Data is contained within the article or Appendix A.

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
