# Peer review of "Enantioselective Synthesis and Pharmacological Evaluation of Aza-CGP37157–Lipoic Acid Hybrids for the Treatment of Alzheimer’s Disease"

_antioxidants, 2022, doi:10.3390/antiox11010112_

Round 1
Reviewer 1 Report
Comments to the authors:
The paper by Angel Cores et al., synthesized and obtained aza-CGP37157-lipoic acid hybrid compounds and shows the effects on cell toxicity, neuroprotection, and anti-inflammatory properties. The introduction is clear. However, in the methods sections, the catalog number of reagents needs to be added. Moreover, some misspellings mistakes need to be fixed.
Major comments
There are misspelling errors that need to be fixed. For example, line 22: compound instead of com,pound; cell line instead of a cellular line or cells; Line 545 add the missing ); Use 0.3 instead of 0,3. In line 431 there are some Spanish words.
Material and Methods.
The authors need to mention whether or not they made the mycoplasma test for the bank/culture cell lines and the passage that they used in all the experiments.
In general, the authors need to add the catalog numbers of all of the reagents that they used for performing the experiments.
The authors need to provide or discuss whether or not the compounds can cross the BBB. For example, do they are planning to perform those experiments?. There are Na/Ca transporter in the cell membrane, these compounds may affect those membrane transporters?.
3.3.1. Line 545: Toxicity was evaluated at increasing concentrations and toxicity was measured as MTT reduction capability. The authors present a table with CC50 information but there is no graphics of toxicity evaluated in a concentration and time-dependent manner. The Authors need to rewrite this phrase or present all the data. Authors need to add the time used for the CC50 experiment, 24 hrs?.
Figures
In general, all the figures need to be briefly described in the figure footnote.
Author Response
Please, see attachment

Reviewer 2 Report
The authors need to address some points in the manuscript before acceptance:
1- Abstract is very brief, authors mentioned only the chemistry part of the work but neglect all biological study. Please present a sufficient description of the activity of prepared compounds and the difference between them as you presented in the conclusion part.
2- some abbreviations need to be spelled out as ALS, ARE, ...
3-In chemical analysis, please add IR and Mass data before NMR
4- the authors need to perform HRESIMS, or add elemental analysis charts to supplementary data.
5- authors need to Rationalize the use of okaidic acid as a method for induction of tau phosphorylation.
6-An important parameter to be measured for oxidative stress is HO-1 as it is highly related to Nrf-2
7- A diagrammatic sketch should be added for more clarification of the experimental design
8- As It was observed, the results of the racemic mixture are very similar to pure compounds, could you explain why you isolate each isomer, could you clarify which is better to preferred biologically. please clarify this point in the conclusion
9- please insert structures of rac-3a and rac-R 3b in figure 2
Author Response
Please, see attachment

Reviewer 3 Report
This study is valuable as a new lead compound for AD. Notably, author found that AZA-CGP 37157 lipoic acid hybrids act neuroprotectively by increasing Nrf2 activity. Although this manuscript is interested, this is not enough and deeply for novel research as regular paper. I propose that to improve and to increase potential impact figures in this field of scientific research.
Major comment
1) Author should be measure Nrf2 nuclear translocation and the Nrf2 expression levels. In addition, author must measure the Nrf2 target molecules. Because the Nrf2-Luc assay alone is not sufficient.
2) Author check the aggregation of beta-Amyloid or tau in SH-SY5Y cells.
3) Author should measure and get the same result except for SH-SY5Y cells. For example CCF‐STTG1, U‐87, and BE(2)‐C cell lines.
Author Response
Please, see attachment

Round 2
Reviewer 2 Report
The authors responded positively to all raised comments.
I think they forgot the addition of HRMS spectra to supplementary data, please add them
Author Response
Reviewer 2 is right, we did not included the HRMS spectra. A new version of supplementary information has been uploaded with HRMS spectra.
Reviewer 3 Report
This manuscript has been improved. Thus, it will be accepted. However, I can't understand which is A, which is B and which is C in Figure 6. This manuscript is accepted if corrected.
Author Response
We checked the uploaded version and reviewer 3 is right, there is no information in figure 6, it must be a problem during uploading of the files. We have uploaded corrected files and checked that all figures are correct.